# Neural Simulation of Actions for Serpentine Robots

**DOI:** 10.3390/biomimetics9070416

**Published:** 2024-07-07

**Authors:** Pietro Morasso

**Affiliations:** Center for Human Technologies Robotics, Brain and Cognitive Sciences Department, Italian Institute of Technology, Via Enrico Melen 83, Bldg B, 16152 Genoa, Italy; pietro.morasso@iit.it

**Keywords:** cognitive robotics, biomimetic robotics, hydrostat, neural simulation of action, prospection, passive motion paradigm, generative body schema, degrees of freedom problem

## Abstract

The neural or mental simulation of actions is a powerful tool for allowing cognitive agents to develop *Prospection Capabilities* that are crucial for learning and memorizing key aspects of challenging skills. In previous studies, we developed an approach based on the animation of the redundant human body schema, based on the Passive Motion Paradigm (PMP). In this paper, we show that this approach can be easily extended to hyper-redundant serpentine robots as well as to hybrid configurations where the serpentine robot is functionally integrated with a traditional skeletal infrastructure. A simulation model is analyzed in detail, showing that it incorporates spatio-temporal features discovered in the biomechanical studies of biological hydrostats, such as the elephant trunk or octopus tentacles. It is proposed that such a generative internal model could be the basis for a cognitive architecture appropriate for serpentine robots, independent of the underlying design and control technologies. Although robotic hydrostats have received a lot of attention in recent decades, the great majority of research activities have been focused on the actuation/sensorial/material technologies that can support the design of hyper-redundant soft/serpentine robots, as well as the related control methodologies. The cognitive level of analysis has been limited to motion planning, without addressing synergy formation and mental time travel. This is what this paper is focused on.

## 1. Introduction

Evolutionary pressure is behind the increase in and expansion of motor redundancy, which is needed in many species to improve their manipulation abilities and creativity. Such bodily evolution has been accompanied by the emergence of specialized brain regions with a dual computational function: control of overt (real) actions and the generation of covert (imagined) actions that allow cognitive agents to reason in a proactive way, in agreement with the fundamental cognitive function of prospection [1].

In the human species, two action categories require the coordination of a very large number of Degrees of Freedom (DoFs): manipulation and phonation. In the former case, the main effector is the hand; in the latter case, it is the vocal tract, which includes the tongue, namely a hydrostat characterized, in principle, by an infinite number of DoFs. In other species that have not developed anything similar to the opposing thumb, a comparable degree of manipulatory ability could be achieved by various types of hydrostats like the elephant trunk [2] or the octopus tentacle [3,4]. Remarkably, with its soft structure and hyper-redundant kinematics, the proboscis can be used for delicate tasks in cluttered and/or unstructured environments, and this feature has quickly captured the interest of robot designers [5,6] in many directions and applications areas [7,8,9]. In particular, a short review of continuum robots [6] clarified the range of architectures, from discrete (nonredundant or mildly redundant) systems to serpentine (hyper-redundant) robots and continuum hydrostats.

The physiology and the biomechanics of elephant trunks [2] and octopus tentacles [3] have been investigated in detail, showing that both species have evolved strategies that reduce the biomechanical complexity of their hydrostat from a functional point of view: elephants and octopuses appear to use strategies similar to vertebrates for transferring an object from one place to another, are also able to reconfigure temporarily the kinematics of the hydrostat into a stiffened, articulated, quasi-jointed structure, and the kinematics of the end effector are consistent with the invariant features of biological motion. 

As stated by Nikolai Bernstein [10], the underlying fundamental problem faced by the human brain is the “degrees of freedom problem”, namely the fact that the highly redundant nature of the body requires a synergy formation process, bridging the biomechanical abundance of the body with the cognitive frugal elegance of the brain. In the human species, this process has been described by Marc Jeannerod [11] as the Neural Simulation of Action, and we propose to extend this approach to serpentine robots like the elephant’s trunk.

From the robotic point of view, a lot of research has been devoted to the actuation/sensorial/material technologies that can support the design of hyper-redundant soft robots. For example, concentric tube robots can adapt to external stimuli and maneuver in complex shapes such as the continuum robot design, which has rolling joints and surface contact joints [12]; tendon/cable mechanisms, where the tension in the cables, when pulled, actuates the intertwining rod skeletons over long distances, are characterized by a relative ease of control and maximization of the propelling force [13]; and origami-inspired continuum robots have attracted great interest in recent years [14], as well as magnetic continuum robots based on an intramolecular polymer complex [15]. The issue of variable stiffness actuators is also important for different approaches, e.g., mechanisms based on S-shaped springs and pneumatic mechanisms [16]. In any case, this is just a small sample of the plethora of recent papers inspired by the snake/serpentine robotic system, with hundreds of them listed in a recent review [17]. As a matter of fact, some systems have gone beyond the proof-of-concept stage and more or less reached the market: for example, the BionicMotion Robot and the BionicSoft Arm by Festo (https://www.festo.com/us/en/e/about-festo/research-and-development/bionic-learning-network/highlights-from-2015-to-2017/bionicsoftarm-id_68209/ (accessed on 3 July 2024)); the FLEX robotic system by Novus (https://novusarge.com (accessed on 3 July 2024)); the i2Snake Robotic Platform for Endoscopic Surgery [18]; the NASA’s EELS architecture [19]; and the OctArm robot [20]. However, the underlying technologies are still evolving, with the main focus being on implementation and low-level control. 

In any case, the potential benefits of robotic hydrostats, including hybrid robotic hydrostats (i.e., robots that combine a traditional multi-link manipulator with a hyper-redundant serpentine end effector), are such that the further development of this kind of technology is likely to occur. On the other hand, the integration of this kind of technology with cognitive capabilities, according to the general requirements of the cyber-physical systems envisioned for the 4th and 5th Industrial revolution [21,22], is still limited and overshadowed by implementation/technical issues. In our opinion, the solution may not come from the adoption of the current Artificial Intelligence methodologies, mainly based on deep/large neural networks, but on an Artificial Cognition formulation, based on an embodied cognition approach [23]. In summary, the aim of this paper is to outline a roadmap for the development of the cognitive capabilities of serpentine robots. In particular, the purpose is to show how prospective capabilities for serpentine robots can be achieved, like the elephant-like robotic architecture depicted in Figure 1. The specific contribution provided by this work consists of extending an approach developed for traditional humanoid robots, namely the Generative Body Schema, based on the Passive Motion Paradigm (PMP) [24,25], with a focus on prospective capabilities. 

Prospection is indeed the crucial function required to cognitively master the physical interaction with the environment. This function has also been characterized as “Mental Time Travel” [26], thus emphasizing the extended role of memory in purposive actions, namely episodic memory and procedural memory, combined in a goal-oriented manner; here, future scenarios are imagined and alternative courses of action are evaluated on the basis of previous experience. The emergence of mental time travel in evolution is probably a crucial step towards the success of the human species, and there is evidence for its presence in other species [26].

The PMP model of synergy formation is a computational implementation of the Neural Simulation Theory [11]; it is based on a force-field approach applied to kinematic networks, namely the idea that multi-joint motor coordination is the consequence of virtual force fields applied to an internal representation of the body, or generative body schema. In the following sections of the paper, it is shown how this model can be extended in a simple, natural way from a conventional skeletal body to a hydrostatic body or a hybrid combination of the two.

## 2. The Generative Extended Body Schema

As suggested in previous papers [27,28,29,30], the basic computational module required for a cognitive agent to achieve prospection is an internal representation of the body or body schema, supported by a unifying simulation/emulation theory of action [31,32,33]. Different from the notion of body image that is mainly a passive representation, the body schema is an active, generative internal model capable of producing spatio-temporal patterns for both real or imagined actions that are consistent with the kinematic–figural invariants that characterize biological motion in humans, primates, as well as elephants. Such invariants are mainly related to the spatio-temporal features of the end effectors rather than joint coordination, suggesting that the brain’s representation of action is more skill-oriented than purely movement/muscle-oriented [34]. 

The basic idea of the PMP model [24,25] is that the coordination of the redundant DoFs of a kinematic chain can be obtained without using ill-posed versions of inverse kinematics but with a series of well-posed transformations: (1) a transformation (via the transpose Jacobian matrix of the kinematic chain) that maps a virtual attractive force field applied to the end effector into the corresponding torque field applied to all the redundant DoFs of the kinematic chain; (2) a transformation that maps the torque field into a vector of joint rotation speeds (via a compliance matrix); and (3) a transformation (via the Jacobian matrix) that maps the joint rotation speeds into the corresponding velocity vector of the end effector. Such a basic PMP model, which is independent of the number of DoFs in the kinematic chain, was later extended [35,36,37] in two directions: (1) a mechanism for protecting the RoM (Range of Motion) of all the joints of the kinematic chain and (2) a mechanism for composing complex gestures from simple primitive gestures in such a way that the kinematic–figural invariance of biological motion is incorporated. As shown in Figure 2, both mechanisms are force-based rather than motion-based, consistent with the principle that in a generative body model, generalized forces are additive whereas generalized motions are not.

In particular, the heart of the body schema is the Jacobian matrix of the end effector *J_e_* and the compliance matrix of the body *C_b_*. The intended motion of the end effector is coded by a force field *F_e_* aimed at a moving target *p_g_* mapped into the corresponding torque field *τ_foc_* by the (transpose) *J_e_* matrix and then into the joint speed rotation vector $\dot{q}$ through the body compliance *C_b_*; finally, the joint rotation vector is mapped into the motion of the end effector via the *J_e_* matrix, thus closing the loop. Such a basic body schema is extended by combining the focal torque field *τ_foc_*, which attracts the end effector to the current target, with a protective RoM field *τ_RoM_* that repulses the joint rotation angles from the joint limits. 

The input to the extended body model is the virtual trajectory *p_g_* of the end effector generated by a composition of Primitive Generators (PGs). As shown in the figure, each PG generates an elastic force field that is characterized by the following parameters: (1) a target point *P*; (2) an initial position $\hat{P}$; (3) the intensity of the elastic force field, modulated by *k_g_*; and (4) a non-linear gating command Γ(t) that is specified by the initial time *t*_0_ and a prescribed duration *T*. By integrating the gated force field over time, a virtual moving target *p*(*t*) that reaches equilibrium at the prescribed time *t_f_ = t*_0_ + *T* and has a bell-shaped velocity profile is generated. The mathematical expression of the activation function Γ(t) used in this study is shown in the Appendix A: it has a sharply growing profile and is capable of forcing the moving target *p*(*t*) to reach equilibrium in the prescribed finite time, thus implementing “terminal attractor” dynamics [38]. The activation command also resets the local integrator to a value $\hat{P}$ that identifies the starting point of the PG.

For generating the complex smooth trajectory of the end effector, it is necessary to instantiate a sequence of PGs, each of them associated with a target point *P_i_*, a starting point ${\hat{P}}_{i-1}$, and an activation command Γ*_i_*(*i*): the starting point of a PG must coincide with the target point of the previous PG. As shown in Figure 2, the force fields generated by the sequence of PGs (*F_g__i_*(*t*)) are linearly combined by the PG composer module, producing a combined force field *F_g_*(*t*) and then the reference trajectory of the end effector *p_g_*(*t*). In a smooth, continuous gesture, the activation functions of subsequent PGs are partially overlapped and the remarkable result is that the spatio-temporal features of the combined gestures intrinsically incorporate the figural–kinematic invariance of biological motion. Thus, the PMP model can tame the abundance of degrees of freedom of the human body by using a small number of primitive force fields: the diffusion of such fields throughout the kinematic network of the internal body model distributes the activity to all the DoFs, with the attractor neurodynamics driven by the instantiated force fields.

In this paper, we show that the PMP-based generative body schema can be extended from the mildly redundant case, typical of the human body, to the hyper-redundant case that includes a trunk-like end effector. Such an extension is simple and quite natural for the intrinsic structure of the PMP-based generative body schema. The heart of the body schema is the Jacobian matrix of the end effector, whose dimensionality can increase as needed without problem. Although the elephant’s trunk is a continuum with an infinite number of DoFs, the practical serpentine implementation will be composed of a large number of equal modules, characterized by a large but finite number of DoFs. As shown in the Appendix A, the overall Jacobian matrix of the elephant’s body includes both the multi-link, mildly redundant part of the body and the hyper-redundant part of the trunk. The primitive generators, the composer and the extended body schema of Figure 2 are formally unchanged by adopting a hydrostatic end effector.

From a computational point of view, the simulation model is an explicit system of first-order differential equations of high dimensionality. The simulation is carried out using Matlab^®^ (MathWorks, Matlab R2023b), adopting the forward Euler method or the 4th order Runge-Kutta method for integrating the differential equation system, with a time step of 0.1 ms. The simulations illustrated in the next section refer to a planar skeleton with 6 DoFs and a planar trunk with 24 DoFs; the complex gesture is composed of nine PGs.

## 3. Results

Figure 3 shows the simulation of a complex gesture performed by the pure trunk-like model, detached from the body skeleton. The planar trunk model has 24 equal modules, each with a 10 cm length and an angular RoM (Range of Motion) of ±30 deg. The simulated gesture is composed of nine PGs: the target point of each PG is marked in the figure by a small, black, open circle; the duration of each PG is 1 s and there is a 50% overlap between subsequent PGs, for a total duration of 5 s. Panel A shows the initial and the final posture, together with the trajectory of the end effector. Panel B also shows the same simulation, with intermediate postures corresponding to instants exhibiting the minimum curvature of the produced trajectory. Both panels were generated with a constant value of 1 for all the 24 elements of the body compliance (*C_b_*). Panel C is different because it is meant to clarify how the cognitive system can manage the predicted presence of an obstacle, depicted as a green filled circle, while attempting to produce the same nine PGs’ gesture with the end effector. The solution is to “freeze” a subset of DoFs, crucial for avoiding the impact with the obstacle, and leave the other DoFs free to rotate inside the specified RoM. This goal is achieved by freezing the compliance if the initial eight DoFs ($C_{b}\left(i\right)=0;i=1\ldots 8$) and setting to 1 the compliance of the remainng DoFs ($C_{b}\left(i\right)=1;i=9\ldots 24$). The picture also suggests that if the obstacle is further shifted down, the gesture would probably need to be aborted.

Figure 4 illustrates the spatio-temporal structure of the trunk motion, corresponding to the nine PGs’ gesture shown in Figure 3. In particular, Panels A and B of Figure 4 correspond to the movement of panel A of Figure 3 where all the elements of the trunk-like kinematic chain have the same unitary compliant value. Panels C and D of Figure 4 correspond to the movement of panel C in Figure 3 where the first eight elements of the trunk-like kinematic chain are frozen. For both the simulated movements, one panel (A and C of Figure 4) shows the wave-like rotation patterns of all the 24 elements of the trunk-like structure: this graph shows that all the rotations are constrained within the prescribed RoM that for this simulation was set to a ±30 deg interval. The satisfaction of the constraint about the RoM is provided by the *τ_RoM_* torque field generated by the RoM protection module of Figure 2. Moreover, comparing panels A and C emphasizes that the rotation patterns of the trunk-like structure are quite different, although the spatio-temporal features of the end effectors are characterized by a kinematic–figural invariance. Such invariance is clarified also in panels B and D of Figure 4, which plot the spatio-temporal patterns of the trajectory depicted by the end effector, namely the speed and the curvature profiles; in both cases, the figure shows that there is a strong inverse correlation between the two profiles that is characteristic of biological motion in humans [28] and in elephants [2]. Remarkably, the figural–kinematic invariance is independent of the number of DoFs recruited for the given gesture. For example, in panel A of Figure 3, all the 24 DoFs of the trunk model are recruited with the same gain, whereas in panel C of Figure 4, the DoFs are frozen: in both cases, the end effector produces the same spatio-temporal patterns.

Figure 5 illustrates the extension of the PMP-based Generative Body Schema to a hybrid model of the elephant body that combines 6 DoFs of the body skeleton and 24 DoFs of the trunk. In panel A, the compliance of the body skeleton is frozen (all the C elements are set to 0) while the C elements of the trunk are equally set to 1. In panel B, the C elements of the body skeleton are mildly unfrozen with growing values from the ankle to the head: C = [0.02 0.04 0.06 0.08 0.10 0.12]. The values of the C elements of the trunk are all set to 1. Both simulations are generated with the same set of PGs that produce a large size gesture, spanning more than one meter vertically. The simulation shows that the coordination of the DoFs of the body skeleton with the DoFs of the trunk improves the smoothness of the gesture.

## 4. Discussion

Animal species that have succeeded in developing hydrostatic manipulators, like elephants and octopuses, exhibit manipulatory dexterity comparable to that of primates, including humans. Such dexterity is necessarily associated with a high degree of intelligence or better well-developed cognitive capabilities that provide the animal with prospective abilities, crucial for mental simulation and mental replay. Such computational capability is considered a crucial component in human learning, in the framework of the neural simulation theory of action that posits mental play/replay as the basic building block of motor cognition, independent of motor control. 

Soft robots and traditional hard robots use different mechanisms to enable dexterous mobility: in the former case, there are distributed deformations with a theoretically infinite number of DoFs, leading to a hyper-redundant configuration space wherein the robot tip can attain every point in the three-dimensional workspace with an infinite number of shapes or configurations. However, continuum deformability is not functionally critical: for example, trunk-like manipulators have been built by using highly redundant rigid structures and electric motors with cable tendons for actuation [39,40].

Although trunk-like soft arms are in principle highly dexterous and adaptable, their performance in terms of payload and spatial movements is limited and requires a very careful design, balancing the influence of key design parameters [41]. This also includes the development and use of architectured structures that indeed are changing the means by which soft robots are designed and fabricated, exploiting material properties, in terms of topology and geometry, that allow their physical and mechanical structural properties to be controlled [42]. In any case, the hyper-redundancy of soft robots implies a sensitivity to the modification of the material properties of the structure, the actuators and the sensors, and this is a critical process of adaptation that can be solved, for example, by means of continual learning techniques [43].

Thus, if soft robotics intends to overcome the proof-of-concept stage, it needs to integrate the actuation–sensing–control level with a cognitive level that allows the robot to “travel in the future” alone or in cooperation with human partners. This aspect of soft robotics has not been investigated so far in a thorough manner, with the exception of different approaches that we may collect under the label of motion planning: approaches based on different forms of inverse kinematic analysis [44,45]; approaches formulated as a least-square optimization problem [46]; approaches related to graph analysis, such as Dijkstra’s algorithm [47], trajectory tracking using an adaptive bounding box [48], or modeling the path planning problem in terms of a rapidly exploring random tree [49]; and approaches based on learning from demonstration using machine learning algorithms [50] or the use of the Actor–Critic method [51].

In conclusion, this paper proposes that the cognitive level of soft/serpentine robots can be formulated as an internal computational model, in terms of a Generative Body Schema based on the Passive Motion Paradigm. This approach does not require inverse transformations and/or explicit optimization processes, whose robustness is difficult to evaluate. Since it is fundamentally based on the combination of force fields, its rationality is based on the equilibrium point hypothesis [52]. As a consequence, the proposed computational model is not limited to motion planning but it covers the more general function of synergy formation that includes the spatio-temporal invariants characteristic of biological motion.

## Figures and Tables

**Figure 1 biomimetics-09-00416-f001:**
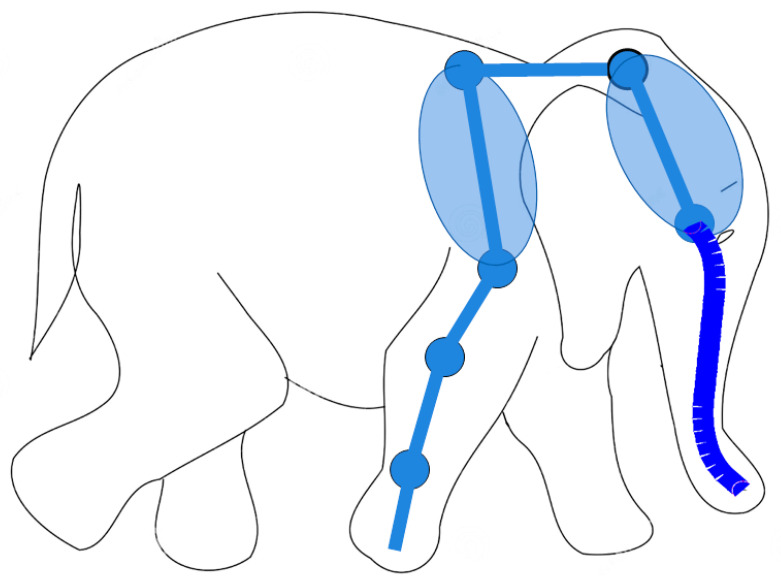
Example of hybrid serpentine robots, with elephant-like robotic architecture that combines a traditional multi-link manipulator (6 DoFs) with a hyper-redundant serpentine end effector (24 DoFs).

**Figure 2 biomimetics-09-00416-f002:**
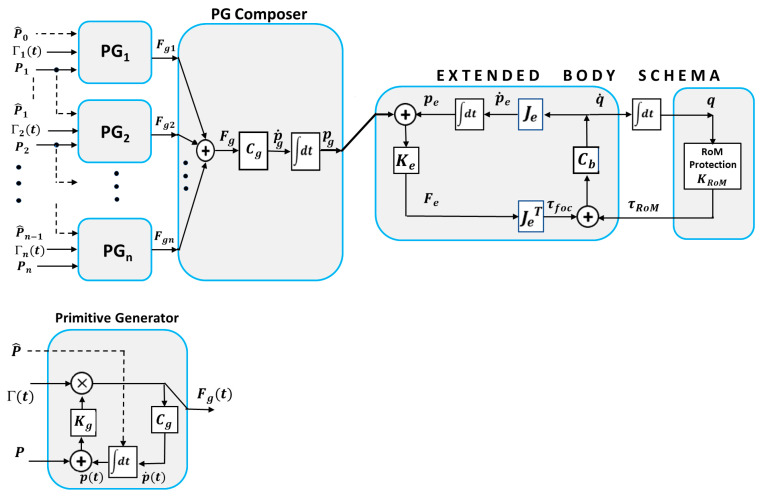
The PMP-based extended body schema.

**Figure 3 biomimetics-09-00416-f003:**
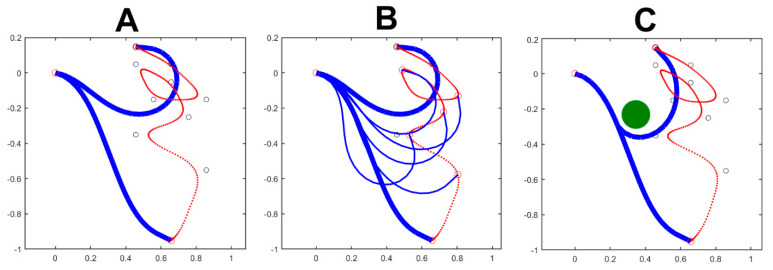
The simulated gesture is composed of nine PGs: the target point of each PG is marked by a small, black, open circle with a 50% overlap between the activation commands of subsequent PGs. Panel (**A**) shows the initial and the final posture, together with the trajectory of the end effector (in red). Panel (**B**) also shows the intermediate postures corresponding to instants exhibiting the minimum curvature of the produced trajectory. Both panels were generated with a constant value of 1 for all the elements of the body compliance (*C_b_*). Panel (**C**) is meant to clarify how the cognitive system can manage the predicted presence of an obstacle, depicted as a green filled circle, while attempting to produce the same nine PGs’ gesture with the end effector. In this panel, a subset of eight DoFs is “frozen”, simply by setting the corresponding compliance equal to 0; the remaining compliance elements are set equally to 1. In the three panels, the measurement units on both axes are m.

**Figure 4 biomimetics-09-00416-f004:**
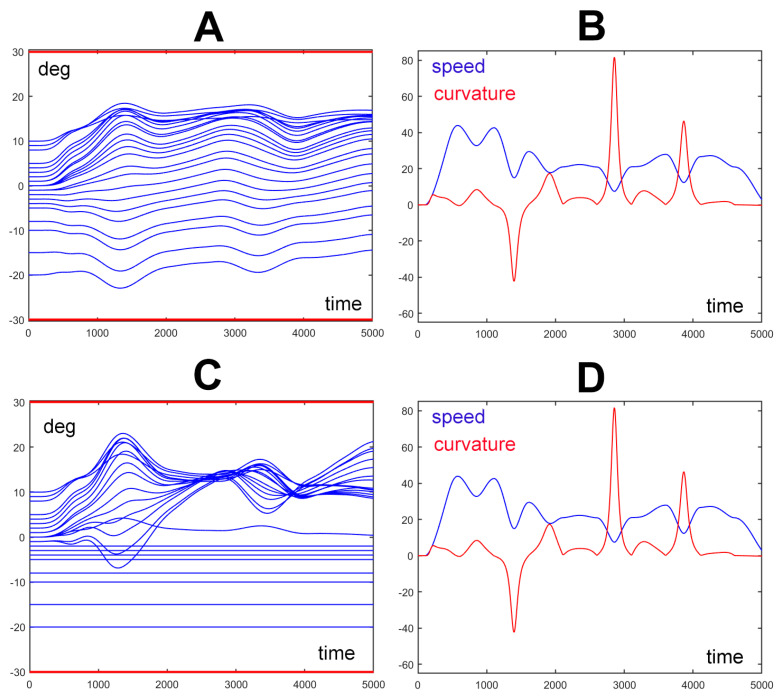
Spatio-temporal structure of the trunk motion for two movements generated by the same set of nine PGs: panels (**A**,**B**) refer to the movement, depicted in Figure 3A, where the compliance elements of all the DoFs of the trunk model are equal; panels (**C**,**D**) refer to the movement, depicted in Figure 3C, where the compliance elements of all the first eight DoFs of the trunk model are frozen (set to 0). Panels (**A**,**C**) show the time plot of the rotation angles of the 24 elements of the trunk model; the two horizontal red lines (at ±30 deg) identify the predefined range of motion of the trunk elements. Panels (**B**,**D**) show the time plot of the speed (in blue: m/s) of the trajectory generated by the end effector and the corresponding curvature (in red: 1/m). For both the simulated movements, the wave-like rotation patterns of the 24 elements of the trunk model are constrained within the prescribed RoM, although the rotation patterns are quite different while the spatio-temporal features of the end effector are invariant.

**Figure 5 biomimetics-09-00416-f005:**
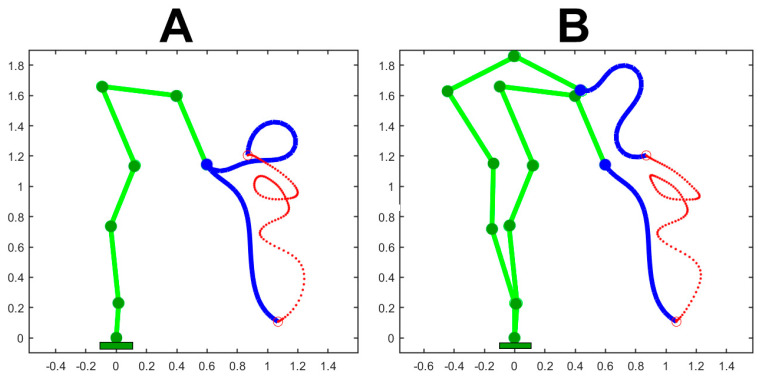
Application of the PMP-based Generative Body Schema to a hybrid model of the elephant body that combines 6 DoFs of the body skeleton and 24 DoFs of the trunk model. In panel (**A**), the compliance of the body skeleton is frozen (all the C elements are set to 0). In panel (**B**), the C elements are mildly unfrozen with growing values from the ankle to the head: C = [0.02 0.04 0.06 0.08 0.10 0.12]. The values of the C elements of the trunk are all set to 1. Both simulations are generated with the same set of PGs.

## Data Availability

Data are contained within the article.

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
