# Peer review of "Neural Simulation of Actions for Serpentine Robots"

_biomimetics, 2024, doi:10.3390/biomimetics9070416_

Round 1
Reviewer 1 Report
Comments and Suggestions for Authors
The manuscript is well-written, and the author has made a significant effort to explain their contribution. However, I have some concerns that I would like to address.
1. The introduction should include the research gap and research questions. Additionally, it should clearly state the author's contribution. Currently, the introduction is very long and vague, failing to clearly address these points.
2. I encourage the author to provide detailed information about the primitive generator and PG composer in Section 2. Instead, Section 2 predominantly discusses PMP in general.
3. The author mentioned that the PMP-based generative body schema could be extended to hyper-redundant cases, including trunk-like end-effectors, but did not provide examples to support this claim. I am particularly interested in the extension to hyper-redundant cases. Please add examples and justify this extension.
4. In Table A1 (page 10), after q=7, the author used "....." and then provided a range of motion (RoM) values (-30,+30) before using "......" again. It is not appropriate to leave rows with empty values. Either leave the entire row empty or provide values for each column.
Overall, the manuscript is clear and well-addressed for the reader. After minor revisions, it can be considered for acceptance.
Author Response
Dear Reviewer,
thank you very much for your careful reading of the manuscript and for your specific comments that I tried to address in the best possible way.
Comment 1. The introduction was modified clarifying the motivation and the specific contribution of the paper (lines 87-93 of the revised manuscript).
Comment 2. Section 2 has been modified explaining in detail the structure and the function of the primitive generator and PG composer (lines 153-158, 163-168 ).
Comment 3. The extension of the PMP-based generative body schema from the conventional redundancy of the human body to the hyper-redundancy of the elephant’s trunk is exemplified in section 3. Consider figures 3 and 5: although they involve potentially a strongly different number of DoFs they were generated exactly with the same script illustrated in figure 1. This point is also addressed in the appendix A1, showing the modular nature of the Jacobian matrix.
Comment 4. The table was modified as suggested.
Reviewer 2 Report
Comments and Suggestions for Authors
This manuscript used a novel Passive Motion Paradigm for hyper-redundant serpentine robots. The simulations effectively demonstrate the algorithm's efficiency and effectiveness, highlighting its potential real-world applications. The research not only addresses a crucial aspect of the neuron model but also contributes valuable new insights into the domain of bio-inspired robots. There are some suggestions that can help to improve the manuscript.
(1) What is the motivation of this work? Understanding the underlying reasons and goals can help clarify the significance and potential impact of the research, especially in advancing current methodologies or addressing existing challenges in the field.
(2) The focus of this paper is on the application of the proposed model to soft robots. Why choose soft robots? What are the advantages of using soft robots? If considering other types of robots, would the model also perform well?
(3) The article is missing a conclusion title, and some punctuation marks are missing. For example, in the abstract section: “In previous studies we developed an approach based on“ should be ”In previous studies, we developed an approach based on”.
Comments on the Quality of English LanguageThere are some grammatical errors or missing punctuation in this article. For example, in the abstract section: “In previous studies we developed an approach based on“ should be ”In previous studies, we developed an approach based on”.
Author Response
Dear Reviewer,
thank you very much for your careful reading of the manuscript and for your specific comments that I tried to address in the best possible way.
Comment 1. The introduction was modified clarifying the motivation and the specific contribution of the paper (lines 87-93 of the revised manuscript).
Comment 2. As a matter of fact, the focus of the paper is on serpentine robots, i.e. hyper-redundant robots with a large but not infinite number of DoFs. The quoted review by Robinson & Davis (1999) on continuum robots clarifies the range of architectures from discrete (nonredundant or mildly redundant) systems to serpentine (hyper-redundant) robots and continuum hydrostats. Although this large range of systems are quite different from the technological/control point of view they are equivalent from the cognitive modeling point of view.
Comment 3. Although there is no specific Conclusion section, the last sentence of the Discussion section (lines 311-318) is formulated as a concluding remark. As regards the missing punctuation marks and the grammatical errors we revised the manuscript in order to eliminate them as much as possible.